# High dose rifampin for 2 months vs standard dose rifampin for 4 months, to treat TB infection: Protocol of a 3-arm randomized trial (2R²)

**Federica Fregonese**[1], **Lika Apriani**[2], **Leila Barss**[3], **Andrea Benedetti**[4,5], **Victoria Cook**[6,7], **Dina Fisher**[3], **Greg J. Fox**[8], **James Johnston**[6], **Richard Long**[9], **Thu Anh Nguyen**[10], **Viet Nhung Nguyen**[11], **Rovina Ruslami**[2], **Dick Menzies**[5,12]*

1 McGill University Health Center-Research Institute, Montreal, QC, Canada, 2 Faculty of Medicine, UNPAD, Bandung, West Java, Indonesia, 3 University of Calgary, Calgary, AB, Canada, 4 McGill University, Montreal, QC, Canada, 5 McGill International TB Centre, Montreal, QC, Canada, 6 British Columbia Centre for Disease Control, Vancouver, BC, Canada, 7 University of British Columbia, Vancouver, BC, Canada, 8 University of Sydney, Sydney, New South Wales, Australia, 9 University of Alberta, Edmonton, AB, Canada, 10 Woolcock Institute, Ha Noi, Vietnam, 11 National TB Program, Ha Noi, Vietnam, 12 McGill University Health Center, Montreal, QC, Canada

* Dick.menzies@mcgill.ca

**Data Availability Statement:** No datasets were generated or analysed during the current study. Full protocol will be made available at the same

## Abstract

### Introduction

Tuberculosis preventive treatment (TPT) is an essential component for TB elimination. In order to be successfully implemented on a large scale, TPT needs to be safe, affordable and widely available in all settings. Short TPT regimens, that are less burdensome than longer regimens, to patients and health systems, are needed. Doses of rifampin higher than the standard 10mg/kg/day were tolerated in studies to reduce duration of treatment for tuberculosis disease (TBD). The objective of this trial is to test the safety of high dose rifampin monotherapy to shorten the duration of the currently recommended TPT of 4 months rifampin.

### Methods and analysis

This is a phase 2b, randomised, controlled, parallel group, superiority, partially-blind trial. Primary outcomes are completion of treatment (as a proxy measure of tolerability) and safety. The two experimental arms comprise 60 days of (i) 20mg/kg/day or (ii) 30mg/kg/day rifampin; the control arm comprises 120 days of 10mg/kg/day rifampin as TPT. Participants are adults and children 10 years or older, eligible for TPT. Completion is the primary outcome, measured by pill count and is defined as taking minimum of 80% of treatment in 120% of allowed time; it will be tested for superiority by logistic regression. Safety outcome comprises proportion of grade 3–5 adverse events and grade 1–2 rash, adjudicated related to study drug, and resulting in permanent drug discontinuation; compared for non-inferiority between each of the two high dose arms and the standard arm, using Poisson regression. A sample size of 1,359 participants will give 80% power to detect a 10% difference in

time of this manuscript publication, in the McGill International TB Center website (https://www.mcgill.ca/tb/projects). All relevant data from this study will be made available upon study completion and publication of study results, in the McGill International TB Center website.

**Funding:** This trial is supported by the Canadian Institute of Health Research (Grant number FDN-143350). The sponsor of this trial is the Research Institute of the McGill University Health Center (MUHC-RI); trial monitoring is done by the coordinating center (study PI and study coordinators), at the MUHC-RI. The funders had no role in study design, data collection and analysis, decision to publish, or preparation of the manuscript.

**Competing interests:** The authors have declared that no competing interests exist.

completion rates and a 1% difference in the safety outcome. The study is conducted in Canada, Indonesia and Vietnam. Enrolment is ongoing at all sites.

## Ethics and dissemination

Approvals from a local research ethics board (REB) have been obtained at all participating sites and by the trial coordinating centre. Approval has been given by drug regulatory agencies in Canada and Indonesia and by Ministry of Health in Vietnam; participants give written informed consent before participation. All data collected are non-nominal. Primary results will be submitted for publication in a peer-reviewed journal when all participants have completed treatment; results of secondary outcomes will be submitted for publication at the end of study; all sites will receive the final data of participants from their sites.

## Trial registration

Trial registered in ClinicalTrials.gov (Identifier: NCT03988933). Coordinating center is the study team working at McGill University Health Center-Research Institute (MUHC-RI); sponsor is the MUHC-RI; funding has been granted by Canadian Institute of Health Research (FDN-143350).

## Introduction

Tuberculosis (TB) preventive treatment (TPT) is a key component in achieving global TB elimination [1] and it is reccomended for close contacts of people with TB, persons living with HIV, and persons with other immune-suppressing conditions at risk of TB infection (TBI) with *M. tuberculosis* [2]. To scale-up TPT in an effort to achieve TB elimination, we need safe and affordable regimens that are acceptable to people in a wide range of settings.

Nine months of isoniazid monotherapy (9H) has demonstrated efficacy [3], but its implementation is limited by its long duration and suboptimal safety profile [4]. Regimens with shorter duration, such as four months of daily rifampin (4R), 12 weeks of weekly rifapentine and isoniazid, or three months of daily rifampin and isoniazid have similar efficacy, better completion and lower health system costs than 9H [5–10]. Additionally, 4R has a better safety profile than 9H [8], but 3–4 months of treatment remains suboptimal for large scale acceptability by people with TB infection and by health systems.

Reducing the duration of TPT, while maintaining safety, tolerability, and efficacy could improve feasibility of utilization; and reduce the treatment burden for patients and health systems. In studies to reduce duration of treatment for TB disease (TBD), increasing the daily dose of rifampin showed promising results [11]. The standard weight-based dose of 10mg/kg rifampin has been increased to 15mg/kg, 20mg/kg and even 30mg/kg/day or more [12–17] with good tolerability and higher proportion of sputum conversion at 8 weeks [18]. We postulate that, similar high doses of rifampin could be used to shorten rifampin monotherapy in TPT. However, there is insufficient evidence to determine which is the optimal dose that will achieve similar efficacy as 4R, without substantial increase in adverse events. In this randomised, phase 2b, controlled, parallel groups, superiority trial, we plan to verify that two months of 20mg/kg and 30mg/kg of rifampin monotherapy are safe and well tolerated, before testing for efficacy in a subsequent phase 3 study. An experimental study in an animal model is ongoing.

## Objectives

The primary objective of this trial is to both compare treatment completion and safety of 4R with two months of rifampin at double (20mg/kg) or triple (30mg/kg) the standard adult rifampin dose weight based. The secondary objective is to compare efficacy, defined as rate of TBD in 26 months following randomization.

## Funding

This trial is supported by the Canadian Institute of Health Research (Grant number FDN-143350). The sponsor of this trial is the Research Institute of the McGill University Health Center (MUHC-RI); trial monitoring is done by the coordinating center (study PI and study coordinators), at the MUHC-RI. Funder and sponsor had no role in any aspect of the study nor on this manuscript.

## Methods

### Interventions

Experimental: 60 daily doses of rifampin 20 mg/kg ($2R_{20}$) or 60 daily doses of 30 mg/kg ($2R_{30}$); Control: 120 daily doses of 10mg/kg rifampin ($4R_{10}$). In all three arms rifampin is self-administered, and doses are based on three weight bands (Table 1).

### Design

This is a three-arm, phase 2b, partially blind, controlled randomized trial. The trial is powered to test superiority of completion and non-inferiority of safety. Randomization is individual and 1:1:1 in the three arms. Members of the same household are randomized to the same arm (i.e., cluster randomized), as long as they are enrolled within 14 days from the date of randomization of the first family member. The study is open label for duration and double blind for dosing of the two high dose arms (300 mg and 450 mg capsules appear identical). The trial has a pharmacokinetics (PK) component, described in more details below.

### Rationale for design features

We expect that completion would be superior in the intervention arms than the comparator arm, due to the shorter duration of therapy. We chose a superiority design, since the rationale to assess these high dose shorter regimens for efficacy would be weak if completion was not improved over 4R.

Blinding in the two experimental arms lowers the risk of bias in reporting symptoms by participants, and in making decisions about drug discontinuation by the treating team. The

**Table 1. Doses of rifampin by weight band.**

| Arm | Blinding* | Target daily dose | Weight bands | | | | | |
|---|---|---|---|---|---|---|---|---|
| | | | 25 to 35kg | | >35kg to 55kg | | >55 kg | |
| | | | Capsules/day | Daily dose | Capsules/day | Daily dose | Capsules/day | Daily dose |
| $4R_{10}$ | Open label | 10 mg/kg | 1x 300mg; | 300 mg | 1x 450mg; | 450 mg | 2x 300mg | 600 mg |
| $2R_{20}$ | Double blind for dose | 20 mg/kg | 2x 300 mg | 600 mg | 3x 300 mg | 900 mg | 4x 300 mg | 1200 mg |
| $2R_{30}$ | Double blind for dose | 30 mg/kg | 2x 450 mg | 900 mg | 3x 450 mg | 1350 mg | 4x 450 mg | 1800 mg |

Note

* 300mg and 450mg rifampin capsules appear identical to maintain blinding.

study is open label for duration since duration is an important determinant of treatment completion.

The randomization is clustered by households to avoid study medication administration errors between family members. The limit of 14 days from the first household member's randomization is to avoid preferential enrollment.

The PK component has been included to assess determinants of variability. There are a few data on PK of higher dose rifampin when used with other medications for treatment of TBD [11]), but less is known on PK of high dose rifampin monotherapy, in persons with Tuberculosis infection (TBI) and in diverse populations (for age, sex, weight, and ethnicity). Higher doses of rifampin have not been used with rifampin monotherapy (for TPT). Therefore, intensive PK with sampling at 6 time points will be performed for 18 participants in each arm at a single study site; while a sparse sampling will be done at all sites for a larger number of participants (450), to ensure representation of diverse populations.

### Assignment of interventions

Randomization is computer generated in blocks of variable size (3, 6 and 9) and stratified by country, and within Canada, by city. In the event of website inaccessibility, every site is provided with sealed opaque envelopes (generated with the same criteria) that are opened following a predefined order if manual randomization is necessary.

### Population and settings

The study is enrolling in 10 TB clinics located in three countries: Indonesia (Bandung), Vietnam (Ha Noi and Ho Chi Minh City) and Canada (Calgary, Edmonton, Montreal, and Vancouver). The list of clinics can be found at clinicaltrials.gov. The clinics are situated in academic hospitals (n = 7), community health centers (n = 2), and university research centers (n = 1).

Participants are adults and children at least 10 years old who have an indication for TPT, according to WHO guidelines for Indonesia and Vietnam, or Canadian guidelines for Canadian sites. Eligible persons must weigh at least 25 kg, have either a positive tuberculin skin test (TST), with the threshold for a positive test defined as per National guidelines, or a positive interferon gamma release assay (IGRA), with TB disease excluded.

Exclusion criteria are pregnancy, liver transaminases (alanine aminotransferase and/or aspartate aminotransferase) 3 times or more higher than upper limit of normal (ULN); grade 3 or 4 abnormalities in hematological tests; prior treatment for TBI or TBD; contraindications to rifampin–due to potential drug interactions that are considered not manageable by treating team or due to history of allergy/ hypersensitivity to rifamycins; and being household contacts of index TB patients with confirmed or presumed rifampin resistant TB. For contacts: index TB patients must have rifampin susceptible TB as per drug susceptibility testing (DST) or GeneXpert result predicting rifampin susceptibility. If neither DST nor GeneXpert are available, then the index TB patients must have no prior TB therapy.

At each site, potential participants who would be recommended TPT by their treating team, are asked to meet the research staff if interested in the study.

Enrollment started at the first study site in September 2019 and is ongoing in all sites.

### Blinding

The treating team and research personnel at site (with exception of study pharmacist) are blinded to dose in the two high-dose arms. Rifampin for high dose arms have been prepared specifically for the study so that capsules of 300mg and 450mg appear identical. At each study

site, once ready to randomize, research staff enter participant's pre-randomization data (see "Data collection") in the trial database, which returns duration of treatment (2 months or 4 months) and a randomization code (if participant is in high dose arms) or daily dose, if participant is in standard dose arm. Pharmacists prepare the bottles for research staff to give to participants in high dose arms with either 300mg or 450mg capsules, without indication of dose on the bottle. In both high dose arms, the daily number of capsules is the same by weight band (Table 1). If blinding needs to be broken for safety reasons, the site contacts the coordinating center, who oversees the unblinding. Interim and final analyses are done by an analyst unaware of the trial-group assignments.

## Outcomes

The primary outcomes are safety and tolerability. Regimen completion is considered a summary measure of tolerability, which combines a participant's decision to stop due to symptoms, inconvenience, or other reasons and the treating teams' decision to stop based on adverse events. Completion is defined as taking at least 80% of doses within 120% of allowed time (i.e. minimally 48 daily doses within 72 days for the high dose arms, and 96 doses within 144 days for the standard arm). The number of doses taken is calculated from the number of pills dispensed and remaining pills counted at each treatment phase follow-up visit.

Safety is defined as the proportion of participants with grade 3 to 5 adverse events, or rash of any grade which resulted in permanent discontinuation of the study drug. If the treating team discontinued the study drug permanently because of concerns of a potential drug-related adverse event, these are evaluated by the adverse events panel. This panel is comprised of 3 members who judge the type, severity, and likely relationship to study drug for each event that resulted in permanent discontinuation of study drug. These judgements are made without knowledge of study arm (blinded), and without knowledge of the judgements of the treating team, nor of the other adverse event panel members. Grading is standardized using American Thoracic Society criteria for hepatotoxicity [19], and National Cancer Institute common criteria for all other types of events [20].

The secondary outcome is efficacy, defined as incidence of TBD in the 26 months post-randomization. Surveillance for incident TB is undertaken actively during treatment phase follow-up visits, and every 3 months post-treatment, up to 26 months after randomization. The final diagnosis of TBD is established by an independent panel of three TB expert physicians, who review all clinical, radiographic, microbiologic and treatment information of persons with possible TB disease, blinded to study arm and independently from each other. TB disease will also be determined passively by each site sending the names of study participants to the local TB authorities to cross-check their list of all persons notified to have TBD during the follow-up period.

## Sample size

Sample size was calculated to detect significantly superior treatment completion of the high dose arms (Table 2). Assuming a rate of completion of 75% with 4R (based on 4R completion in previous trials [8, 21] we need to enroll 412 participants per arm to detect a 10% better completion rate. Allowing for a 10% withdrawal, or otherwise not analyzable participants, we need 453 per arm, or a total of 1359 participants (Table 2).

For the non-inferiority analysis of safety, the sample size estimations are presented in Table 3. If the proportion of grade 3–5 adverse events considered possibly/probably related to study drug and resulting in permanent treatment discontinuation is 1% or 2% in standard arm- as seen in previous trials [8, 21]—and there is 1% greater occurrence in high dose arms,

**Table 2. Sample size required to detect superior completion of $2R_{20}$ or $2R_{30}$ compared to $4R_{10}$ with 80% power and α = 0.05.**

| Completion rate | | N per arm to detect significant difference[*] | | | |
|---|---|---|---|---|---|
| $4R_{10}$ | $2R_{20}$ or $2R_{30}$ | No clustering | All Clustering | Mixed clustering and not | Total N (3 arms) |
| 70% | 75% | 1248 | 2484 | 2075 | 6226 |
| | 80% | 290 | 578 | 483 | 1450 |
| | 85% | 118 | 235 | 196 | 589 |
| **75%** | 80% | 1091 | 2172 | 1815 | 5446 |
| | **85%** | **247** | **493** | **412** | **1237** |
| | 90% | 97 | 193 | 151 | 453 |
| 80% | 85% | 903 | 1797 | 1505 | 4515 |
| | 90% | 196 | 391 | 306 | 917 |
| | 92.5% | 115 | 230 | 180 | 540 |

[*]**Assumptions**: (i) 65% of all enrolled subjects will be clustering in same household, and 35% will be other risk groups, as was seen in previous trial (8); (ii) The intra-class correlation coefficient (ICC) or clustering effect of households on completion will be 0.33 –as was demonstrated among study subjects who had at least one other family member in previous trial (8); (iii) Use 4 as estimated average number of household contacts, based on two systematic reviews [22, 23].

then the sample size needed to address completion will also be sufficient to establish non-inferiority for safety, as total needed will be between 702 and 1158 (Table 3). If the adverse event in the higher dose arms is of 2% greater occurrence than standard arm (i.e. is 3% in high dose vs 1% in standard or 4% vs 2%) or if the rate in standard arm is 4%, then we would not have the number needed to conclude non-inferiority for safety.

Strategies to enroll expected sample size, are specific to each site and reviewed with coordinating center.

## Statistical analysis

**Primary analyses.** The proportion completing treatment in each high dose arm will be compared against the standard arm, and tested for significance with logistic regression, using an identity link, and estimated via generalized estimating equations (GEE) -to account for clustering by household. An exchangeable correlation structure and empirical standard errors

**Table 3. Sample size required to conclude non-inferiority (maximum tolerated difference = 4%) of $2R_{20}$, or $2R_{30}$ in terms of grade 3–5 events compared to $4R_{10}$ with a power of 80% and α = 0.05.**

| Grade 3–5 Adverse event rate | | Number required per arm to conclude non-inferiority[*] | |
|---|---|---|---|
| $4R_{10}$ | $2R_{20}$ or $2R_{30}$ | N per arm | Total N–for 3 arms |
| **1%** | **2%** | **234** | **702** |
| | 3% | 698 | 2094 |
| | 4% | 3468 | 10,404 |
| **2%** | **3%** | **386** | **1158** |
| | 4% | 1035 | 3105 |
| | 5% | 4805 | 14,415 |
| 3% | 4% | 534 | 1602 |
| | 5% | 1366 | 4098 |
| | 6% | 6114 | 18432 |

[*] **Assumptions**: Cluster effect (or Intraclass correlation coefficient ICC) in terms of severe adverse event of 0.05; 4 contacts per household. In this table, for simplicity, all participants are assumed to be household contacts–since the ICC is low and has very modest effect on sample size required

will be used. Analysis of treatment completion will be done as modified intention to treat (MITT), i.e., on all participants enrolled, with exclusion only of participants who were excluded post-randomization, for ineligibility, following procedures defined in the protocol for these exclusions.

The proportion of participants with grade 3–5 adverse events during treatment (and up to 30 days after treatment interruption for drug-drug interactions), or grade 1–2 rash with permanent treatment discontinuation adjudicated as possibly or probably related to the study treatment by the adverse events panel, will be compared between each of the two high dose arms and the standard arm. Adjustment for clustering will be done by Poisson regression (as these adverse events should be relatively rare outcomes), including an offset of log(person time), using GEE, with an exchangeable correlation structure and empirical standard errors. To assess non-inferiority of safety, we will use the 95% confidence interval approach, and compare the upper limit of the difference versus a margin of 4%. This margin was chosen as safety is essential for a TPT regimen. In previous trials, standard dose rifampin had 2% of these type of adverse events [8, 21], therefore a maximum adverse events rate of 6% (4% higher) was seen as the maximum that would be acceptable to patients and providers. Analysis of safety will be done on all enrolled participants who took at least one dose of study treatment.

**Secondary analyses.** First, lab testing is done routinely at 2 weeks for the two high dose arms, and again at 4 weeks, whereas this testing is only done at 4 weeks for the standard arm, which may bias detection of adverse events. Hence, we plan a secondary analysis for completion and safety that will exclude treatment interruptions and adverse events which occurred only because of the routine lab testing done at 2 weeks in the high dose arms. Second, analysis of non-inferiority of treatment completion, with maximum allowable difference of 5% and one-sided significance level.

Last, the incidence of TBD (microbiologically confirmed and all forms) in the 26 months post-randomization per 100 person-years of follow-up, will be compared between all three arms. This analysis will be done as MITT and as per-protocol (i.e., in all participants enrolled and who completed treatment they were assigned to).

All these analyses will include adjustment for clustering by household. In stratified analysis, results will be presented by indication for TPT. Sensitivity analyses will be conducted whereby analysis are stratified by study centre and by country.

## Data collection

**Pre-randomization.** Potential participants who consent to participate undergo medical evaluation and blood tests to confirm eligibility. Clinical and demographic information are collected. Documentation of a positive TST or IGRA must be available; TBD must be excluded by chest X-ray (CXR), symptom questionnaire and medical examination for all participants. Microbiological tests are required for participants with abnormalities on CXR or symptoms compatible with TB. Women of childbearing potential have a pregnancy test done.

**Treatment phase.** All arms have 3 follow-up visits (at different times) during the treatment period (Fig 1).

At each treatment phase visit participants are examined and questioned regarding symptoms of TBD and of possible adverse events. Pill counts are done by research personnel, and pills for the next interval dispensed. Routine blood tests include hemogram (hemoglobin, platelet counts, and leukocytes), liver transaminases, bilirubin, creatinine, and blood urea nitrogen. These are taken after 4 weeks of treatment in all arms, and also at 2 weeks for high dose arms only (Fig 1), to ensure safety in the experimental arms. Additional visits and lab tests can be done if needed, as required by the site medical team.

| | STUDY PERIOD | | | | | | | | |
|---|---|---|---|---|---|---|---|---|---|
| | Enrollment | Allocation | Post-allocation | | | | Post intervention | | Close |
| TIMEPOINT | -6 months to time 0 (included) | Time 0 | + 2 weeks | + 4 weeks | +8 weeks | +16 weeks | + 5 months | Every 3 months | + 26 months |
| **ENROLMENT:** | | | | | | | | | |
| **Eligibility screen** | X | | | | | | | | |
| **Informed consent** | X | | | | | | | | |
| *Evaluation pre-randomization* | X | | | | | | | | |
| **Randomization** | | X | | | | | | | |
| **INTERVENTIONS:** | | | | | | | | | |
| *High dose (2 arms)* | | | | | | | | | |
| *Standard dose* | | | | | | | | | |
| **ASSESSMENTS:** | | | | | | | | | |
| *TST/ IGRA; CXR* | X | X | | | | | | | |
| *Medical history & assessment** | X | X | | | | | | | |
| *Baseline lab tests*** | X | | | | | | | | |
| *Sampling for population PK* | | | | X | | | | | |
| *HIGH DOSE ARMS:* Pill count | | | X | X | X | | | | |
| Symptoms' questionnaire | | | X | X | X | | X | X | X |
| Lab tests for safety | | | X | X | | | | | |
| *STANDARD ARM:* Pill count | | | | X | X | X | | | |
| Symptoms' questionnaire | | | X*** | X | X | X | X | X | X |
| Lab tests for safety | | | | X | | | | | |

**Fig 1. SPIRIT figure for schedule of enrollment, interventions, and assessments. NOTES**: * Medical history includes: previous TB or LTBI; indications for TPT; concomitant conditions and treatments, allergies, contraception (for women of childbearing potential), current symptoms. Baseline medical assessment includes: height, weight, medical evaluation. ** Lab tests are baseline are: Complete blood count (White blood cells, platelets, hemoglobin, hematocrit); liver transaminase (ALT, or AST); bilirubin; pregnancy test for women of childbearing potential; HIV test (may be offered by treating team).For participants with HIV: CD4 and viral load. ***only symptoms questionnaire is done at 2 weeks for standard arm (not a complete FU visits.

For PK component: at week 4, a minimum of 450 participants (all arms combined), at all sites, have two blood samples collected at 2 and 4 hours after reported drug intake, and, in a subsample of 27 adults ($\geq$18 years old) - 9 per arm- and 27 children (10 to <18years old) -9 per arm- a series of venous blood samples are collected just before, and at 1, 2, 4, 8, and 12 h after witnessed drug intake, at one study site.

Due to required restrictions to reduce risk of COVID-19 infections during pandemic, some parts of pre-randomization and treatment-phase visits, including consent process, enrollment, pill count and symptoms questionnaires, may be done remotely, following procedures approved by local ethics committees at each site. Required lab tests and imaging recommendations remain unchanged during public health restrictions related to COVID-19. Where needed, study medications are dispensed by courier, after preparation by the site pharmacist, preserving blinding.

To minimize differences in the visit frequency and intensity of follow-up which may affect treatment completion, total number and approximate spacing of visits are the same for all arms. Participants who decide to stop therapy against medical advice are encouraged to restart and complete therapy by their treating team. Participants who maintain the decision to stop, are offered alternative TPT outside of the study or may decide not to take TPT.

**Post-treatment phase.** All participants who stop study treatment for any reason, have post treatment follow-up telephone calls every 3 months until 26 months post randomization. Every effort is made to minimize loss to follow-up. At each follow-up call, participants will be questioned regarding current symptoms of TBD, any intercurrent diagnoses of TBD, or hospitalization for any reason. If TBD is presumed (at any time), this is investigated and managed

by treating team with assistance of research staff. All presumed TBD are reported to coordinating center, and evaluated by an independent panel of three expert TB clinicians (see above, in "Outcomes").

## Data management

Data collected at all study timepoints are recorded in specifically designed paper case report forms (CRF), and then entered in the study website. Data verification of data for a random sample of participants is done by the coordinating center periodically throughout the study, against source documents at the sites for laboratory and imagining results reported, as well as for paper CRF completed at site versus data entered in website. Data management of data for all participants is done before interim and final analysis.

## Monitoring

The study is conducted in compliance with this protocol and the International Conference on Harmonisation—Good Clinical Practice (ICH-GCP) in all sites. In each site, regulatory requirements of the Country health authority (Health Canada for Canadian sites, National Agency of Drug and Food Control- Badan POM-, in Indonesia) and of the Ministry of Health in Vietnam, are followed.

To ensure high quality data, all staff at participating sites receive initial and periodic training in the study protocol, procedures, and data requirements. Research staff have certification of training in Good Clinical Practice.

Study sites receive monitoring visits every 6 months (or more often if needed) by the coordinating center (study PI and study coordinator/s) to ensure that the study is conducted in accordance with protocol and to check for quality of data collected. When travel restrictions apply, monitoring visits are done remotely.

During monitoring visits from the coordinating center, and if any visits occur for auditing, REB review and regulatory inspections, study site investigators provide direct access to all essential documents including source documents. Independent auditing may occur during study, by sponsor (RI-MUHC), funding agency (CIHR) or regulatory agency (HC).

## Ethical considerations

**Safety of intervention.** Rifampin is a safe and well-known drug, used for more than 45 years. Minor adverse effects are explained to participants at enrollment. Serious adverse events, such as hepatitis, are rare and usually resolve once treatment is stopped. Visits and lab tests during treatment follow-up are in place to capture possible events and act upon them before they become severe. As rifampin can interact with other medications, monitoring of potential interactions is discussed with participants and done at follow-up visits. Alternatives to study treatment (including other TPT) are discussed with participants and offered by treating teams when study treatment is stopped.

**Monitoring of safety.** This trial has a Data Safety and Monitoring Board (DSMB), which is composed by the three TB expert physicians (Dr. Mike Lauzardo; Dr. Rick O'Brien and Dr. Randall Reves). The DSMB is responsible to review the two planned interim safety analyses and to review on an ad-hoc and immediate basis any serious or grade 3–4 adverse events that are unexpected and considered treatment related, as well as any deaths that occur during treatment phase.

Unexpected serious adverse events related to study drug are reported to local research ethics board (REB), to Health Canada, and to all sites.

There are two planned interim analyses, after a total of 150 and 450 participants (i.e., 50 and 150 per arm respectively) have completed treatment phase. The DSMB review the results and give recommendation on trial continuation, need for modification or to stop enrolment if necessary. They also recommend if further interim analyses are needed.

Participants may voluntarily withdraw from the study at any time without prejudice to ongoing or future treatment from the treating team. Management of all participants who decide to withdraw from the study, is discussed and decided by participant and their treating team.

**Confidentiality.**   Data collected in the trial are non-nominal. Participants are assigned a unique study identifier (ID) and all information sent to the coordinating centre, or reviewed by independent panels, contain only this study ID.

Personal information is stored only at the sites, double locked, in a secure location, and safe-guarded by the site PI. Study documents are stored for 25 years after completion of the study.

At end of study: in Canada, lists of participants' names will be forwarded by registered cou-rier to provincial health authorities to assess if they developed TBD, following procedures approved by provincial privacy commissions. In Indonesia and Vietnam, the participants' names will be cross-checked against reported TB cases at National level, using appropriate safeguards for participant confidentiality.

**Ethics review and consent.**   All participants provide signed informed consent before ran-domization. Children aged 10–17 sign an assent form, in addition to parental signed consent. Consent is obtained by research personnel. Protocol, consent, and assent forms have received approval by the REB of the MUHC and by research ethics committees/boards at all participat-ing centres (see Table 4). Letter of Non-Objection has been received by Health Canada for Canadian sites; approval from Badan POM for Indonesian sites and from Ministry of Health for Vietnam sites. In Canadian sites, when consent is obtained remotely due to COVID restric-tions, the process follows the instructions from local REB.

**Trial oversight.**   The Scientific Advisory Committee (SAC) meets twice per year and pro-vides recommendations regarding scientific aspects of the study including study design, inter-ventions, and outcomes. The Data Safety and Monitoring Board (DSMB) is described above. The Trial steering committee is composed of the study PI, one investigator per site, the study statistician, and the study coordinator. This committee reviews the recommendations of the DSMB and the SAC, as well the overall progress of the trial, and need for study amendments. It is responsible for all the decisions regarding stopping enrolment to any study arm, or the study. If such a decision is taken, the research ethics committees at all participating sites are notified.

## Dissemination

The results pertaining the primary outcomes of the trial (i.e., completion and safety) will be submitted as a manuscript to a peer reviewed journal, as soon as possible once the treatment phase of the study is completed. Results of analyses for secondary outcomes (efficacy) and of secondary analysis or sub-analyses, will be subsequently submitted to peer reviewed journals. Site investigators can request data from the coordinating center for other secondary analyses—to be submitted to peer reviewed journals. Authorship eligibility will be discussed for each publication with the trial steering committee; there will be no use of professional writers.

A database relative to participants of that site will be sent to investigator/s of each site, once the final data management has been done. Full protocol will be made available at the same time of this manuscript publication, in the McGill International TB Center website (https://www.mcgill.ca/tb/projects); data generated from the study will be made publicly available at the time of publication of study results, in the same website.

**Table 4. World Health Organization trial registration data set.**

| Data category | Information |
|---|---|
| Primary registry and trial identifying number | ClinicalTrials.gov: NCT03988933 |
| Date of registration in primary registry | 18 June 2019 |
| Secondary identifying numbers | MUHC-REB: 2R2/2019-5360; |
| | University of Alberta: Pro00093849; |
| | University of British Columbia: H19-01058 |
| | Health Canada NOL: HC6-24-c228350 (original), last amendment 246554. |
| | Universitas Padjadjaran Research Ethic Committee:1391/UN6.KEP/EC/2019. |
| | University of Calgary Conjoint Health Research Ethics Board: REB19-1179; National Lung Hospital (Vietnam) IRB: 547/2020/NCLH. |
| Source(s) of monetary or material support | Canadian Institute of Health Research (FDN-143350). |
| Primary sponsor | Research Institute of McGill University Health Centre (RI-MUHC) |
| Secondary sponsor(s) | Dick Menzies (study PI) and his team at RI-MUHC (study coordinators) |
| Contact for public queries | Dick Menzies, MD, dick.menzies@mcgil.ca, |
| | Tel: 514-934-1934 ext 32128 |
| | 5252 de Maisonneuve West, Room 3D.58 |
| | Montreal, Quebec, Canada H4A 3S5 |
| Contact for scientific queries | Dick Menzies, MD, dick.menzies@mcgil.ca, |
| | Tel: 514-934-1934 ext 32128 |
| | 5252 de Maisonneuve West, Room 3D.58 |
| | Montreal, Quebec, Canada H4A 3S5 |
| Public title | High dose versus standard dose rifampin to treat TB infection. |
| Scientific title | High dose rifampin for 2 months vs standard dose rifampin for 4 months, to treat TB infection: protocol of a 3-arm randomized trial ($2R^2$). |
| Countries of recruitment | Canada, Indonesia, Vietnam. |
| Health condition(s) or problem(s) studied | Tuberculosis preventive treatment, tuberculosis infection, rifampin, high dose rifampin. |
| Intervention(s) | Intervention comparator: (i) rifampin 20mg/kg/day for 60 days; (ii) rifampin 30mg/kg/day for 60 days. Capsules of 300mg or 450mg rifampin, looking exactly the same, have been prepared for the interventional arms of the study; dosage is based on weight bands. |
| | Control comparator: rifampin; 10mg/kg/day for 120 days (open label) |
| Key inclusion and exclusion criteria | Ages eligible for study: ≥10 years |
| | Sexes eligible for study: both |
| | Accepts healthy volunteers: no |
| | Inclusion criteria: ≥ 10 years with indication for tuberculosis (TB) preventive treatment and proof of TB infection (i.e. TST or IGRA positive). |
| | Exclusion criteria: history of allergy/ hypersensitivity to rifamycins; concomitant treatment with potential drug-drug interactions that are considered not manageable by treating team; pregnancy; liver transaminases ≥3 times upper limit of normal (ULN); grade 3 or 4 abnormalities in hematological tests; prior treatment for TBI or TBD; being household contacts of TB patients with confirmed or suspected rifampin resistant TB; weigh <25Kg. |

*(Continued)*

**Table 4.** (Continued)

| Data category | Information |
|---|---|
| Study type | Interventional; |
| | Allocation: randomized. Randomization sequence is generated in blocks of variable size (3, 6 and 9) and stratified by country, and within Canada, by city, by a program embedded in the study website. At enrolment, research team at each site enters the information for a potential new participant and the website returns duration of treatment (4 months or 2 months) and if, in interventional arms, a code corresponding to 20mg/kg or 30mg/kg/day. Ten codes are utilised for each interventional arm. Unblind pharmacists at each sites prepare the medication corresponding to codes for interventional arms (double blind); or the medication for the control arm (open label). |
| | Intervention model: parallel assignment. |
| | Masking: partially blind: i.e. double bind (for participants, caregiver, investigator, outcomes assessors) in intervention arms; and open label for control arm. |
| | Primary purpose: treatment of TB infection. |
| | Phase: IIb |
| Date of first enrolment | September 2019 |
| Target sample size | 1359 |
| Recruitment status | Recruiting |
| Primary outcome(s) | 1.Treatment completion: defined as taking at least 80% of doses within 120% of allowed time (i.e. minimally 48 daily doses within 72 days for the interventional arms, and 96 doses within 144 days for the comparator arm); |
| | 2. Safety: defined as proportion of participants with grade 3 to 5 adverse events, or rash of any grade which resulted in permanent discontinuation of the study drug. |
| Key secondary outcomes | Efficacy: Incidence of TBD in the 26 months post-randomization |
| Ethics Review | Approved, by each site's Ethic Committee: |
| | Calgary: Conjoint Health Research Ethics Board (CHREB), University of Calgary; 23 Sep 2019. |
| | Edmonton: University of Alberta Health Research Ethics Board—Biomedical Panel; 08 November 2019; |
| | Montreal: MUHC-REB; 17 July 2019; |
| | Indonesia: Universitas Padjadjaran Research Ethic Committee; 20 November 2019; |
| | Vancouver: University of British Columbia Clinical Research Ethics Board; 10 September 2019; |
| | Vietnam Ministry of Health IRB: 28 December 2020; National Lung Hospital IRB (Vietnam): 23 April 2020. |
| Completion date | Not competed yet (expected completion of recruitment by end of 2022; expected completion of study follow-up end of 2024). |
| Summary Results | Not yet available (study still ongoing) |
| IPD sharing statement | Plan for data sharing will be finalized once study is completed and detailed in main publication. |

## Discussion

Achieving the WHO TB elimination goals requires a short, safe and effective TPT regimen. In addition, for TPT to be implemented by health care systems in all settings, it needs to be widely available and inexpensive. Among the currently recommended drugs for TPT [2], rifampin is relatively safe, well known, affordable and accessible (as licensed in all countries). When used at the current standard dose though, while very safe, it still needs to be taken for 4 months.

Higher doses (20mg/kg; 30mg/kg) of rifampin are safe and well tolerated when used in TBD treatment [12, 13, 15, 16] and could be the way to reduce duration of TPT, while maintaining rifampin monotherapy safety profile. This study's main goal is to demonstrate that high dose rifampin is safe and well tolerated when used for TBI. Doses for which this can be demonstrated, can then be studied in an efficacy trial.

A shorter treatment with completion comparable or superior to current standard treatment, could have potential benefit because of fewer doses and less time to be in follow-up, which may be convenient for both patients and programs. A treatment with rifampin monotherapy, that can be self-administered with relatively few capsules (up to 4) per day for 60 days, can be of benefit to the large population of people at risk of TBD and to health systems in all settings.

## COI

In Canadian sites: the cost of medications for the standard arm were covered by each Province (as $4R_{10}$ is the standard of care in these sites). Alberta and British Columbia covered also the cost for high dose arms for their participants. For high dose arms, commercially available 300mg rifampin (Rofact) has been re-compounded by Pharmacie Linda Frayne (Montreal, Canada), into 300mg and 450mg capsules, looking the same. For Vietnam sites, 300mg and 450mg capsules made for the study for high dose arms were bought from Svizera Europe B.V. (the Netherlands), 150mg and 300mg rifampin capsules for standard arm were purchased from Mekopharm (Vietnam). For Indonesia, medications for all arms were bought from Indofarma (Indonesia). None of these companies have any role in study design or conduct, nor have they provided any funding or in kind support.

The trial web site and randomisation program has been developed by the staff of the Biomedical Telematic Laboratory of the Quebec Respiratory Health research Network, Sherbrooke, QC.

## Supporting information

**S1 Checklist. SPIRIT fillable checklist completed.**
(DOC)

**S1 File. 2R2 protocol approved.**
(DOCX)

**S2 File. 2R2 informed consent form.**
(DOCX)

**S3 File. McGill University Health Center-Research Ethic Board- Initial approval.**
(PDF)

**S4 File. McGill University Health Center-Research Ethic Board- Approval current version.**
(PDF)

**S5 File. 2R2 ethical approval—Indonesia site.**
(PDF)

**S6 File. 2R2 ministry of health ethical committee approval Vietnam 28 Dec 2020.**
(PDF)

**S7 File. 2R2 national lung hospital approval letter- Vietnam.**
(PDF)

**S8 File. Questionnaire on inclusivity in global health research.**
(DOCX)

## Acknowledgments

**DSMB** members: Dr.Mike Lauzardo; Dr. Rick O'Brien and Dr.Randall Reves; **SAC** members: Dr. Bill Burman, Dr. Andy Vernon, Dr. Ben Marais, Dr. Olivia Oxlade; **Coordinating center**: Chantal Valiquette; Lisandra Lannes; *Biomedical Telematic Laboratory of the Quebec Respiratory Health research Network*, Sherbrooke, QC: Eric Rousseau, Yvan Fortier and Mina Dilgui; **Active TB panel**: Dr. Sharyn Mannix; Dr.Barry Rabonovitch; Dr. Marcel Behr.

## Author Contributions

**Conceptualization:** Dick Menzies.

**Formal analysis:** Andrea Benedetti.

**Funding acquisition:** Dick Menzies.

**Investigation:** Federica Fregonese, Lika Apriani, Leila Barss, Victoria Cook, Dina Fisher, Greg J. Fox, James Johnston, Richard Long, Thu Anh Nguyen, Rovina Ruslami, Dick Menzies.

**Methodology:** Federica Fregonese, Lika Apriani, Leila Barss, Victoria Cook, Dina Fisher, Greg J. Fox, James Johnston, Richard Long, Thu Anh Nguyen, Rovina Ruslami, Dick Menzies.

**Project administration:** Federica Fregonese, Dick Menzies.

**Supervision:** Lika Apriani, Leila Barss, Victoria Cook, Dina Fisher, Greg J. Fox, James Johnston, Richard Long, Thu Anh Nguyen, Viet Nhung Nguyen, Rovina Ruslami, Dick Menzies.

**Writing – original draft:** Federica Fregonese, Dick Menzies.

**Writing – review & editing:** Federica Fregonese, Lika Apriani, Leila Barss, Andrea Benedetti, Victoria Cook, Dina Fisher, Greg J. Fox, James Johnston, Richard Long, Thu Anh Nguyen, Viet Nhung Nguyen, Rovina Ruslami, Dick Menzies.

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
