## [Decision Letter · Decision Letter 0]

29 Sep 2022

PONE-D-22-16478High dose rifampin for 2 months vs standard dose rifampin for 4 months, to treat TB infection: protocol of a 3-arm randomized trial (2R2).PLOS ONE

Dear Dr. Menzies,

Thank you for submitting your manuscript to PLOS ONE. After careful consideration, we feel that it has merit but does not fully meet PLOS ONE’s publication criteria as it currently stands. Therefore, we invite you to submit a revised version of the manuscript that addresses the points raised during the review process.

Your manuscript has been assessed by a panel of expert reviewers, whose comments are appended below. The reviewers, while broadly positive about your protocol, have highlighted some concerns about aspects of the methodology and rationale for the study design used. Please ensure you respond to each point carefully in your response to reviewers document, and modify your manuscript accordingly.

We look forward to receiving your revised manuscript.

Kind regards,

Joseph Donlan

Senior Editor

PLOS ONE

Journal Requirements:

6. We note that the original protocol file you uploaded contains a confidentiality notice indicating that the protocol may not be shared publicly or be published. Please note, however, that the PLOS Editorial Policy requires that the original protocol be published alongside your manuscript in the event of acceptance. Please note that should your paper be accepted, all content including the protocol will be published under the Creative Commons Attribution (CC BY) 4.0 license, which means that it will be freely available online, and any third party is permitted to access, download, copy, distribute, and use these materials in any way, even commercially, with proper attribution.

Therefore, we ask that you please seek permission from the study sponsor or body imposing the restriction on sharing this document to publish this protocol under CC BY 4.0 if your work is accepted. We kindly ask that you upload a formal statement signed by an institutional representative clarifying whether you will be able to comply with this policy. Additionally, please upload a clean copy of the protocol with the confidentiality notice (and any copyrighted institutional logos or signatures) removed.

Reviewers' comments:

Reviewer's Responses to Questions

**Comments to the Author**

1. Does the manuscript provide a valid rationale for the proposed study, with clearly identified and justified research questions?

Reviewer #1: Yes

Reviewer #2: Yes

Reviewer #3: Yes

Reviewer #4: Yes

2. Is the protocol technically sound and planned in a manner that will lead to a meaningful outcome and allow testing the stated hypotheses?

Reviewer #1: Yes

Reviewer #2: Yes

Reviewer #3: Yes

Reviewer #4: Yes

3. Is the methodology feasible and described in sufficient detail to allow the work to be replicable?

Reviewer #1: Yes

Reviewer #2: Yes

Reviewer #3: Yes

Reviewer #4: Yes

4. Have the authors described where all data underlying the findings will be made available when the study is complete?

Reviewer #1: Yes

Reviewer #2: No

Reviewer #3: Yes

Reviewer #4: Yes

5. Is the manuscript presented in an intelligible fashion and written in standard English?

Reviewer #1: Yes

Reviewer #2: Yes

Reviewer #3: No

Reviewer #4: Yes

6. Review Comments to the Author

You may also provide optional suggestions and comments to authors that they might find helpful in planning their study.

Reviewer #1: A clear and well presented manuscript. Clearly articulates study design and will be an important contribution to the field. No specific additional comments.

Reviewer #2: Review: High dose rifampin for 2 months vs standard dose rifampin for 4 months, to treat TB infection: protocol of a 3-arm randomized trial (2R2).

This manuscript is a protocol to evaluate the adherence to, and safety of, a shorter, high-dose rifampin regimen compared to a standard-dose rifampin tuberculosis preventive treatment (TPT) regimen, with a secondary aim to evaluate efficacy over a 26-month period. The protocol is well written and clear. The reviewer has a few comments. In addition, for publication purposes, the reviewer thinks that consistent terminology is important (see suggestions in comments), that all abbreviations should first be explained (and once abbreviated, used consistently).

Major comments:

1. The authors use “tuberculosis infection (TBI)” and “latent TB” interchangeably, as well as TB disease (TBD) and “active TB”. The reviewer thinks that for this manuscript it would be better to consistently use “TB infection (TBI)” and “TB disease” (with or without abbreviation “TBD”) throughout, as these refer to the same entities. (see e.g., line 188-189; suggest use “TBI or TBD”)

2. Line 187: For the exclusion criteria of liver transaminases raised >3 times normal – is this for any one (ALT or AST) or both together (supposedly the first option). Suggest add: “…liver transaminases (alanine aminotransferase and/or aspartate aminotransferase)…”

3. Lines 210-211: The reviewer cannot see a reason for unblinding for safety reasons, as if there is a safety issue the patient is likely to discontinue rifampin and not be continued on another rifampin regimen in the trial? Unblinding always creates a risk for bias.

4. Lines 307-308: What is the purpose of analysis of non-inferiority of treatment completion? Superiority is what is aimed for, as without superiority of treatment completion, there is no real purpose for the shorter regimen?

5. Line 309: “Last, the incidence of active TB (microbiologically confirmed and all forms) in…” Suggest “TB disease”, but does this only refer to microbiologically confirmed TB – children are also included and in them it may be more difficult to microbiologically confirm TB? What does “all forms” refer to – pulmonary and extrapulmonary TB?

6. Lines 321-322: Are patients not clinically examined to exclude extrapulmonary TB?

7. Line 336: “two blood samples collected at 2 and 4 hours after reported drug intake,” Timing of sampling after drug intake is very important to compare concentrations (for PK) - how is the time of taking the medication going to be verified?

8. Line 337: Suggest define age of children for PK study (10-<18 years?)

9. Lines 351-352: If participants opt out of the study and receive alternative TPT, do they remain in study follow up or are they excluded from further follow-up (as the other TPT regimen could bias outcome)?

10: Line 393: Drug-drug interactions are a specific problem with rifampin and several antiretroviral medications, but no mention is made in the protocol regarding inclusion or exclusion of people living with HIV. Could the authors clarify?

11: Lines 474-483: The different manufacturers of rifampicin and different formulations used may have an important effect on both safety and PK - how is this going to be evaluated?

Minor comments/suggested corrections.

Abstract:

- lines 58 and 72: Suggest start sentence with Upper case letter as in line 50 after “:”

- line 61: delete “:” after “control arm”

- line 67: To the reviewer’s knowledge, two connected words followed by a noun should be written with a hyphen: in this case “high-dose arm” and “standard-dose arm” – these terms should then consistently used throughout the manuscript.

- line 75: “informed” (not “inform”)

Introduction:

- line 94: “4 months of” (add “s” to month)

- Objectives: line 115: suggest add: “adult rifampin dose” to read “standard adult rifampin dose”, as it is higher in children

- Funding: line 120: “…by the Canadian…” (can use “a”, but then need to add “grant” as part of the sentence in addition to the part in brackets)

Methods:

- Table 1: In first column, both “bind” words should be “blind”. In addition, I think it would be good to add a footnote to the table explaining that the high-dose rifampin 300 and 450 mg capsules look the same. The reason is that this explanation only comes much later in the protocol, and the reviewer’s first thought was how can this be blinded if there are different strength formulations which in normal practice would not be the same.

- line 143: Suggest change to: “…, described in more detail below.”

- line 143: Suggest add in brackets: “…and double blind for dosing of the two high-dose arms (300 mg and 450 mg capsules look the same).” Same reason as above

- line 153: should be “decisions”

- line 157: should be: “…first household member’s randomization.” (apostrophe s moved)

- lines 160-162: Pharmacokinetcs (PK) already abbreviated in line 160 therefore in lines 161 and 162 (and further) “PK” should be used (or alternatively do not abbreviate the word)

- line 161: “rifampin” used throughout except here?

- line 163: TBI has not been abbreviated before – write out in full

- line 164: rifampin (lower case “r”)

- line 171: suggest change “:” to “,”

- line 191: The word “suspected” has fallen into disfavor regarding TB. Mostly replaced with “presumed”

- line 192: The reviewer thinks that using “susceptible” and “susceptibility” is better than using “sensitive” and “sensitivity” (see line 193), because with statistics “sensitivity” is also used

- Line 204: “at each study site” (not site study)

- line 218: Change to “team’s”

- line 224: “…as the proportion…”

- line 227: “adverse events” have been used many times already, but here suddenly abbreviated without explanation – should be consistent in using abbreviations

- lines 235, 238, 240, 242: suggest use “TB disease” (or TBD if abbreviated before) instead of “active TB”

- line 262: Suggest add “for safety” to read: “…be sufficient to establish non-inferiority for safety,…” – same in line 266 at the end of the sentence (for the reader not to get confused)

- line 263: “real adverse events” (multiple?)

- Table 3: Explain abbreviations used in the footnote (ICC, HHC)

- line 282: spaces “(GEE) – to account”

- line 297: “these types of adverse events” – add “adverse”

- line 333: lab tests (plural)

- line 343: “ethics”

- Lies 358, 359, 360: TB disease rather than active TB; replace “suspected” with “considered” or “presumed”

- line 378: Write out “GCP”

- line 390: Change “side effects” to “adverse effects”; Also: “Serious adverse events, such as…”

- line 406: Should be “DSMB”

- line 419: If it is at the end of the study, then this should be "developed TB disease"?

- line 448: “analyses – to” (add spaces)

Reviewer #3: This is a very nice protocol of a study that is already registered and ongoing. It is well described, and the full protocol etc. look to be attached as supplementary material.

Comments / questions from me are minor:

1. Lines 92-97 outline current LTBI treatment regimens which are helpful background. Would it be good to mention the 1RpH regimen (https://www.nejm.org/doi/full/10.1056/nejmoa1806808) which was published in NEJM in this list?

2. One thing I pondered is why you needed to do this 'safety / tolerability' study before progressing to a full Phase III study of this dose of rifampicin for LTBI. The emergent data (now up to 40+mg/kg, https://pubmed.ncbi.nlm.nih.gov/33542056/) consistently suggest that 30mg/kg is well below the limit of tolerability for this drug and I wonder if you might just have proceeded to an efficacy trial, with embedded secondary safety endpoints, in order to find the answer to the main question that you really have more quickly? This, of course, is not a criticism - just a query and something I'd like to have seen more justification of. Will it even be possible to "roll on" into a full Phase III study, with extended follow-up of the patients you already have, if this study is successful?

3. The superiority design is great, but do you really need to show anything more than non-inferiority here to justify the value of the new regimens? If taking less medicine is as easy and safe (and eventually effective) as more medicine that feels like reason enough not to give extra (unnecessary) pills). From a patient perspective non-inferiority would seem to be enough and I'm not sure if I really agree that "the rationale to

assess these high dose shorter regimens for efficacy would be weak if completion was not

improved over 4R". If everyone completed (irrespective of 2 vs 4 months) the 2 month regimen wouldn't be superior but it would still be worth shortening treatment for all because less medicine is enough. The reason this might matter is the possibility that, during a trial adherence and completion of therapy might be 'artificially' improved vs. routine practice (a Hawthorn type effect), and that this 'artificial' improvement might be exaggerated on the more difficult / control arm. Is there a risk that you are disappointed by, and discard, a useful approach to shorter LTBI therapy because you fail to find hard-to-prove superiority of the new regimen under trial conditions?

4. The partial blind doesn't seem to allow any blinding between the control (10mg/kg) arm and the experimental arms (different number of tablets in the control arm). I can see the practical reasons for that, but wonder if it does slightly confound the main comparison with the control? I guess not, and lots of high-dose rifampicin studies are completely open label for similar reasons but did you consider any possible issues here?

5. Spelling errors in row 3 and 4 of Table 1 (should be "blind" not "bind").

These comments are really discussion points prompted by reading the manuscript - I have no doubt that this is a useful study, which the TB community will be interested to read about (and will eagerly await the results of).

Reviewer #4: Abstract, introduction :The objective of this trial is to test the safety of high dose rifampin monotherapy to shorten the duration of the currently recommended 4 months rifampin. Do you mean for prevention of TB?

Line 87: typo in reccomeded

Otherwise, good luck!

7. PLOS authors have the option to publish the peer review history of their article (what does this mean?). If published, this will include your full peer review and any attached files.

Reviewer #1: No

Reviewer #2: No

Reviewer #3: **Yes: **Dr Derek J Sloan

Reviewer #4: No

---

## [Author Response · Author response to Decision Letter 0]

4 Oct 2022

Response to Reviewers’ and Editor’s requests for revisions 

2R2 Protocol manuscript - PONE-D-22-16478

Part I: Response to Reviewers:

Comments to the Author

1. Does the manuscript provide a valid rationale for the proposed study, with clearly identified and justified research questions?

Reviewer #1: Yes

Reviewer #2: Yes

Reviewer #3: Yes

Reviewer #4: Yes

2. Is the protocol technically sound and planned in a manner that will lead to a meaningful outcome and allow testing the stated hypotheses?

Reviewer #1: Yes

Reviewer #2: Yes

Reviewer #3: Yes

Reviewer #4: Yes

3. Is the methodology feasible and described in sufficient detail to allow the work to be replicable?

Reviewer #1: Yes

Reviewer #2: Yes

Reviewer #3: Yes

Reviewer #4: Yes

4. Have the authors described where all data underlying the findings will be made available when the study is complete?

Reviewer #1: Yes

Reviewer #2: No

Reviewer #3: Yes

Reviewer #4: Yes

A: We revised the sentence relative to data sharing and added information regarding where the data will be found. This revision is on page 22 of the manuscript with track changes and reads: “Full protocol will be made available at the same time of this manuscript publication, in the McGill International TB center website (https://www.mcgill.ca/tb/projects); data generated from the study will be made publicly available at the time of publication of study results, in the same website.” We hope this has addressed this issue.

5. Is the manuscript presented in an intelligible fashion and written in standard English?

Reviewer #1: Yes

Reviewer #2: Yes

Reviewer #3: No

Reviewer #4: Yes

A: We have addressed all specific comments by the Reviewers below. Hopefully we have now addressed this concern.

6. Review Comments to the Author

You may also provide optional suggestions and comments to authors that they might find helpful in planning their study.

Reviewer #1: A clear and well presented manuscript. Clearly articulates study design and will be an important contribution to the field. No specific additional comments.

Reviewer #2: Review: High dose rifampin for 2 months vs standard dose rifampin for 4 months, to treat TB infection: protocol of a 3-arm randomized trial (2R2).

This manuscript is a protocol to evaluate the adherence to, and safety of, a shorter, high-dose rifampin regimen compared to a standard-dose rifampin tuberculosis preventive treatment (TPT) regimen, with a secondary aim to evaluate efficacy over a 26-month period. The protocol is well written and clear. The reviewer has a few comments. In addition, for publication purposes, the reviewer thinks that consistent terminology is important (see suggestions in comments), that all abbreviations should first be explained (and once abbreviated, used consistently).

Major comments:

1. The authors use “tuberculosis infection (TBI)” and “latent TB” interchangeably, as well as TB disease (TBD) and “active TB”. The reviewer thinks that for this manuscript it would be better to consistently use “TB infection (TBI)” and “TB disease” (with or without abbreviation “TBD”) throughout, as these refer to the same entities. (see e.g., line 188-189; suggest use “TBI or TBD”)

A: The terminology has been corrected to be TBI and TBD throughout.

2. Line 187: For the exclusion criteria of liver transaminases raised >3 times normal – is this for any one (ALT or AST) or both together (supposedly the first option). Suggest add: “…liver transaminases (alanine aminotransferase and/or aspartate aminotransferase)…”

A: The reviewer is correct – our intention is to exclude if EITHER is abnormal. Now corrected, the exclusion criterion reads: […] liver transaminases (alanine aminotransferase and/or aspartate aminotransferase) 3 times or more higher than upper limit of normal (ULN); […]

3. Lines 210-211: The reviewer cannot see a reason for unblinding for safety reasons, as if there is a safety issue the patient is likely to discontinue rifampin and not be continued on another rifampin regimen in the trial? Unblinding always creates a risk for bias.

A: We agree that safety reasons for unblinding in this trial are very unlikely to present. The study medication in all arms is the same (i.e. rifampin), and even in the event of a suspected Adverse Drug Reaction, knowing the exact dosage is very unlikely to change the medical conduct (i.e. discontinuing the medication). Nevertheless, we were required to have a procedure for unblinding, in case needed, by the regulatory agency and ethics committee reviewers. 

4. Lines 307-308: What is the purpose of analysis of non-inferiority of treatment completion? Superiority is what is aimed for, as without superiority of treatment completion, there is no real purpose for the shorter regimen?

A: See our response to a related question by Reviewer 3, who seems to have the opposite opinion about superiority vs non-inferiority. We believe that, even if a shorter treatment was not superior but “just” non-inferior to the standard treatment, there could be potential benefit for patients or programs of having fewer doses, and less time to be in follow-up. We also think it likely that TB programs would prefer shorter regimens, that are 'not worse’ then the standard one for completion, to reduce costs of follow-up. This point, since it seems more controversial than we first thought, has been added to the Discussion.

5. Line 309: “Last, the incidence of active TB (microbiologically confirmed and all forms) in…” Suggest “TB disease”, but does this only refer to microbiologically confirmed TB – children are also included and in them it may be more difficult to microbiologically confirm TB? What does “all forms” refer to – pulmonary and extrapulmonary TB?

A: All forms – will include extra-pulmonary and clinically diagnosed TB disease. As regards terminology, and as explained above: we have used now TB disease instead of active TB.

6. Lines 321-322: Are patients not clinically examined to exclude extrapulmonary TB?

A: Potential participants are evaluated for possible TB disease, including physical examination, by treating physicians. They decide on further investigations to exclude TB disease, based on participant’s medical history, and symptoms. The protocol requires chest-X ray, symptoms questionnaire and medical examination for all. This has been now specified on line 322.

7. Line 336: “two blood samples collected at 2 and 4 hours after reported drug intake,” Timing of sampling after drug intake is very important to compare concentrations (for PK) - how is the time of taking the medication going to be verified?

A: In this trial, all doses are self-administered, which we believe is an important feature of the design as it will make the regimen much more feasible for later implementation. At the time of PK sampling, participants are asked at what time they took study medication. This will be only two hours later, so we believe their recall should be accurate, and we cannot see any reason to think there will be bias in reporting, so any minor inaccuracies will represent “random misclassification”. Also, participants are called the day before the PK sampling to be reminded that they need to retain hour at which medication is taken. 

8. Line 337: Suggest define age of children for PK study (10-<18 years?)

A: this specification has been added, the line now reads: “… in a subsample of 27 adults (≥18 years old) - 9 per arm- and 27 children ( 10 to <18years old) -9 per arm-...”

9. Lines 351-352: If participants opt out of the study and receive alternative TPT, do they remain in study follow up or are they excluded from further follow-up (as the other TPT regimen could bias outcome)?

A: All participants, regardless of completion of treatment (or alternate treatment), are in post-treatment follow-up, for up to 26 months from enrollment. Information about alternative TPT treatment and related outcomes (completion, non-completion, adverse events) are collected during post-treatment follow-up visits. Patients who take alternate treatment will be included in the intention to treat (per randomization) analysis but excluded from the efficacy (per protocol) analysis.

10: Line 393: Drug-drug interactions are a specific problem with rifampin and several antiretroviral medications, but no mention is made in the protocol regarding inclusion or exclusion of people living with HIV. Could the authors clarify?

A: People with HIV are included, as this group of patients have potential for large benefits of TPT. However, the antiretroviral therapy they are taking at enrollment must be compatible with concomitant use of rifampin, as is the case for other concomitant treatments. The study protocol includes a table with the most common interaction between antiretrovirals and rifampin.

11: Lines 474-483: The different manufacturers of rifampicin and different formulations used may have an important effect on both safety and PK - how is this going to be evaluated?

A: This is a valid point, which has been discussed during protocol development and trial implementation. A sample of medications used in all arms at each site will be collected and analysed for content of active ingredient by a centralized lab. This analysis is part of the Statistical Analysis Plan. 

Minor comments/suggested corrections.

Abstract:

- lines 58 and 72: Suggest start sentence with Upper case letter as in line 50 after “:”

- line 61: delete “:” after “control arm”

A: Both corrected, thanks 

- line 67: To the reviewer’s knowledge, two connected words followed by a noun should be written with a hyphen: in this case “high-dose arm” and “standard-dose arm” – these terms should then consistently used throughout the manuscript.

A: We are not experts in grammar but prefer to keep as separate words.

- line 75: “informed” (not “inform”)

A: corrected, thanks 

Introduction:

- line 94: “4 months of” (add “s” to month)

A: corrected, thanks 

- Objectives: line 115: suggest add: “adult rifampin dose” to read “standard adult rifampin dose”, as it is higher in children

A: corrected, thanks 

- Funding: line 120: “…by the Canadian…” (can use “a”, but then need to add “grant” as part of the sentence in addition to the part in brackets)

A: corrected, thanks 

Methods:

- Table 1: In first column, both “bind” words should be “blind”. In addition, I think it would be good to add a footnote to the table explaining that the high-dose rifampin 300 and 450 mg capsules look the same. The reason is that this explanation only comes much later in the protocol, and the reviewer’s first thought was how can this be blinded if there are different strength formulations which in normal practice would not be the same.

A: The correction has been done and the following note added to Table 1: “300mg and 450mg rifampin capsules appear identical to maintain blinding.” 

- line 143: Suggest change to: “…, described in more detail below.”

A: corrected, thanks 

- line 143: Suggest add in brackets: “…and double blind for dosing of the two high-dose arms (300 mg and 450 mg capsules look the same).” Same reason as above

A: Explanation has been added.

- line 153: should be “decisions”

- line 157: should be: “…first household member’s randomization.” (apostrophe s moved)

A: Both corrected, thanks 

- lines 160-162: Pharmacokinetcs (PK) already abbreviated in line 160 therefore in lines 161 and 162 (and further) “PK” should be used (or alternatively do not abbreviate the word)

A: Change was made to use PK, thanks 

- line 161: “rifampin” used throughout except here?

A: rifampicin corrected to be rifampin, for consistency

- line 163: TBI has not been abbreviated before – write out in full

A: TBI has been spelled out.

- line 164: rifampin (lower case “r”)

- line 171: suggest change “:” to “,”

A: Both done

- line 191: The word “suspected” has fallen into disfavor regarding TB. Mostly replaced with “presumed”

A: Change made.

- line 192: The reviewer thinks that using “susceptible” and “susceptibility” is better than using “sensitive” and “sensitivity” (see line 193), because with statistics “sensitivity” is also used

A: Change made.

- Line 204: “at each study site” (not site study)

- line 218: Change to “team’s”

- line 224: “…as the proportion…”

A: The above three have been changed 

- line 227: “adverse events” have been used many times already, but here suddenly abbreviated without explanation – should be consistent in using abbreviations

A: We have eliminated the AE abbreviation.

- lines 235, 238, 240, 242: suggest use “TB disease” (or TBD if abbreviated before) instead of “active TB”

A: Corrected.

- line 262: Suggest add “for safety” to read: “…be sufficient to establish non-inferiority for safety,…” – same in line 266 at the end of the sentence (for the reader not to get confused)

A: “For safety” was added in both sentences. 

- line 263: “real adverse events” (multiple?)

A: The word “real” has been removed to increase clarity. The sentence now reads: “If the adverse event in the higher dose arms is of 2% greater occurrence than standard arm (i.e. is 3% in high dose vs 1% in standard or 4% vs 2%) or if the rate in standard arm is 4%, then we would not have the number needed to conclude non-inferiority for safety.”

- Table 3: Explain abbreviations used in the footnote (ICC, HHC)

A: These abbreviations have been explained or spelled out.

- line 282: spaces “(GEE) – to account”

- line 297: “these types of adverse events” – add “adverse”

- line 333: lab tests (plural)

- line 343: “ethics”

A: The four above have been corrected.

- Lies 358, 359, 360: TB disease rather than active TB; replace “suspected” with “considered” or “presumed”

- line 378: Write out “GCP”

- line 390: Change “side effects” to “adverse effects”; Also: “Serious adverse events, such as…”

- line 406: Should be “DSMB”

- line 419: If it is at the end of the study, then this should be "developed TB disease"?

- line 448: “analyses – to” (add spaces)

A: All corrected.

Reviewer #3: This is a very nice protocol of a study that is already registered and ongoing. It is well described, and the full protocol etc. look to be attached as supplementary material.

Comments / questions from me are minor:

1. Lines 92-97 outline current LTBI treatment regimens which are helpful background. Would it be good to mention the 1RpH regimen (https://www.nejm.org/doi/full/10.1056/nejmoa1806808) which was published in NEJM in this list?

A: At time of protocol development 1HP was not yet among recommended treatments, so it is not in the trial protocol. Up to now it this regimen has only been tested in HIV coinfected population, we opted to not include this in the paper. 

2. One thing I pondered is why you needed to do this 'safety / tolerability' study before progressing to a full Phase III study of this dose of rifampicin for LTBI. The emergent data (now up to 40+mg/kg, https://pubmed.ncbi.nlm.nih.gov/33542056/) consistently suggest that 30mg/kg is well below the limit of tolerability for this drug and I wonder if you might just have proceeded to an efficacy trial, with embedded secondary safety endpoints, in order to find the answer to the main question that you really have more quickly? This, of course, is not a criticism - just a query and something I'd like to have seen more justification of. Will it even be possible to "roll on" into a full Phase III study, with extended follow-up of the patients you already have, if this study is successful?

A: We agree with reviewer there is data regarding tolerability of higher doses of rifampin, but in patients with TB disease. We think it is important to establish safety and tolerability of high doses in the population with TB infection. Tolerance for safety is different for preventive treatment, such as TPT, than for treatment of a life-threatening disease, such as TBD. Also, tolerability for adverse events (even minor), in the TBI population, with no symptoms and an otherwise healthy life, may differ from what has been reported in patients with TBD. Finally high doses of rifampin, in previous TBD studies has not been used as monotherapy, but in combination with other anti-TB medications, and this may have a role both on safety and tolerability. For these reasons we think a safety trial is necessary before a phase 3 trial. At the same time, since we plan to conduct a phase 3 trial, if this phase 2B trial is successful, we have followed participants for two years after end of TPT, for the secondary objective of efficacy. This will allow data from participants of this trial to be included in the future phase 3 trial, reducing the time and sample size needed to test efficacy. 

3. The superiority design is great, but do you really need to show anything more than non-inferiority here to justify the value of the new regimens? If taking less medicine is as easy and safe (and eventually effective) as more medicine that feels like reason enough not to give extra (unnecessary) pills). From a patient perspective non-inferiority would seem to be enough and I'm not sure if I really agree that "the rationale to assess these high dose shorter regimens for efficacy would be weak if completion was not improved over 4R". If everyone completed (irrespective of 2 vs 4 months) the 2 month regimen wouldn't be superior but it would still be worth shortening treatment for all because less medicine is enough. The reason this might matter is the possibility that, during a trial adherence and completion of therapy might be 'artificially' improved vs. routine practice (a Hawthorn type effect), and that this 'artificial' improvement might be exaggerated on the more difficult / control arm. Is there a risk that you are disappointed by, and discard, a useful approach to shorter LTBI therapy because you fail to find hard-to-prove superiority of the new regimen under trial conditions?

A: See our response to Reviewer 1, who appears to have the opposite opinion. We have a secondary analysis which will look at non-inferiority of completion, and the overall results of primary and secondary analyses will be considered when discussion on a phase 3 trial will be held.

4. The partial blind doesn't seem to allow any blinding between the control (10mg/kg) arm and the experimental arms (different number of tablets in the control arm). I can see the practical reasons for that, but wonder if it does slightly confound the main comparison with the control? I guess not, and lots of high-dose rifampicin studies are completely open label for similar reasons but did you consider any possible issues here?

A: We agree that comparison with the standard arm could be biased given the open label design. This is why safety will be evaluated by an independent adverse event panel, which adjudicates the type, grade and relationship to study drug, while blinded to dose and also to duration. The final analysis will be conducted by a biostatistician who is blinded to all arms as well. 

5. Spelling errors in row 3 and 4 of Table 1 (should be "blind" not "bind").

A: Corrected

These comments are really discussion points prompted by reading the manuscript - I have no doubt that this is a useful study, which the TB community will be interested to read about (and will eagerly await the results of).

Reviewer #4: Abstract, introduction :The objective of this trial is to test the safety of high dose rifampin monotherapy to shorten the duration of the currently recommended 4 months rifampin. Do you mean for prevention of TB?

A: We have modified the sentence to add clarity: “The objective of this trial is to test the safety of high dose rifampin monotherapy to shorten the duration of the currently recommended TPT of 4 months rifampin.” 

Line 87: typo in reccomeded

A: Thanks for spotting this, it has been corrected. 

Otherwise, good luck!

7. PLOS authors have the option to publish the peer review history of their article (what does this mean?). If published, this will include your full peer review and any attached files.

Do you want your identity to be public for this peer review? For information about this choice, including consent withdrawal, please see our Privacy Policy.

Reviewer #1: No

Reviewer #2: No

Reviewer #3: Yes: Dr Derek J Sloan

Reviewer #4: No

Part II: Response to Editor's comments:

A: We have edited the file format to meet these requirements.

A: The questionnaire on inclusivity in global research has been uploaded as Supporting Information (S9. Questionnaire on inclusivity in global health research).

A: List of authors and affiliations have been edited as per journal’s guidelines

A: Ethic statement appears under Methods section of manuscript (and Methods section of Abstract). 

A: the following captions for Supporting information were added at the end of the manuscript.

S1. SPIRIT fillable checklist completed

S2. 2R2 Protocol approved. 

S3. 2R2 Informed Consent Form

S4. McGill University Health Center-Research Ethic Board- Initial approval

S5. McGill University Health Center-Research Ethic Board- Approval current version

S6. 2R2 Ethical approval - Indonesia site

S7.2R2 Ministry of Health Ethical Committee approval Vietnam 28 Dec 2020

S8. 2R2 National Lung Hospital approval letter- Vietnam

S9. Questionnaire on inclusivity in global health research.

6. We note that the original protocol file you uploaded contains a confidentiality notice indicating that the protocol may not be shared publicly or be published. Please note, however, that the PLOS Editorial Policy requires that the original protocol be published alongside your manuscript in the event of acceptance. Please note that should your paper be accepted, all content including the protocol will be published under the Creative Commons Attribution (CC BY) 4.0 license, which means that it will be freely available online, and any third party is permitted to access, download, copy, distribute, and use these materials in any way, even commercially, with proper attribution.

Therefore, we ask that you please seek permission from the study sponsor or body imposing the restriction on sharing this document to publish this protocol under CC BY 4.0 if your work is accepted. We kindly ask that you upload a formal statement signed by an institutional representative clarifying whether you will be able to comply with this policy. Additionally, please upload a clean copy of the protocol with the confidentiality notice (and any copyrighted institutional logos or signatures) removed.

A: We agree to publicly share the protocol at the time of publication of this manuscript. Please find now submitted as S2 a complete protocol, without confidentiality notice. As this is an investigator-initiated trial and the authors are also acting as coordinating center for the study, please let us know if a formal statement from the institutional representative is still needed. 

A: Reference list has been checked. No revisions were made.

---

## [Editor Report · Decision Letter 1]

10 Nov 2022

High dose rifampin for 2 months vs standard dose rifampin for 4 months, to treat TB infection: protocol of a 3-arm randomized trial (2R2).

PONE-D-22-16478R1

Dear Dr. Menzies,

I participated as a reviewer for the initial evaluation of this manuscript, and was then invited by the journal to serve as academic editor for this paper.

We’re pleased to inform you that your manuscript has been judged scientifically suitable for publication and will be formally accepted for publication once it meets all outstanding technical requirements.

Kind regards,

Derek J. Sloan, PhD

Guest Editor

PLOS ONE

Additional Editor Comments (optional):

Reviewers' comments:

No additional comments following last review - thank you for your thoughtful responses to our prior questions and discussion points.

---

## [Editor Report · Acceptance letter]

23 Nov 2022

PONE-D-22-16478R1 

High dose rifampin for 2 months vs standard dose rifampin for 4 months, to treat TB infection: protocol of a 3-arm randomized trial (2R^2^). 

Dear Dr. Menzies:

I'm pleased to inform you that your manuscript has been deemed suitable for publication in PLOS ONE. Congratulations! Your manuscript is now with our production department. 

Kind regards, 

on behalf of

Dr. Derek J. Sloan 

Guest Editor

PLOS ONE